

# TurkMedNLI: a Turkish medical natural language inference dataset through large language model based translation

İskender Ülgen Oğul[1], Fatih Soygazi[2] and Belgin Ergenç Bostanoğlu[1]

[1] Computer Engineering, Izmir Institute of Technology, İzmir, Turkey
[2] Computer Engineering, Adnan Menderes University, Aydın, Turkey

## ABSTRACT

Natural language inference (NLI) is a subfield of natural language processing (NLP) that aims to identify the contextual relationship between premise and hypothesis sentences. While high-resource languages like English benefit from robust and rich NLI datasets, creating similar datasets for low-resource languages is challenging due to the cost and complexity of manual annotation. Although translation of existing datasets offers a practical solution, direct translation of domain-specific datasets presents unique challenges, particularly in handling abbreviations, metric conversions, and cultural alignment. This study introduces a pipeline for translating a medical NLI dataset into Turkish, which is a low-resource language. Our approach employs fine-tuning the Llama-3.1 model with selected samples from the Medical Abbreviation dataset (MeDAL) to extract and resolve medical abbreviations. Consequently, NLI pairs are refined with extracted abbreviations and subjected to metric correction. Later, the processed sentences are then translated using Facebook's No Language Left Behind (NLLB) translation model. To ensure quality, we conducted comprehensive evaluations using both machine learning models and medical expert review. Our results show that BERTurk achieved 75.17% accuracy on TurkMedNLI test data and 76.30% on the normalized test set, while BioBERTurk demonstrated comparable performance with 75.59% accuracy on test data and 72.29% on the normalized dataset. Medical experts further validated the translations through manual assessment of sampled sentences. This work demonstrates the effectiveness of large language models in adapting domain-specific datasets for low-resource languages, establishing a foundation for future research in multilingual biomedical NLP.

# INTRODUCTION

Natural language inference (NLI) is a fundamental task in natural language processing (NLP) that involves determining the relationship between two sentences named premise and hypothesis. These relationships are typically classified as entailment, contradiction, or neutrality. These classes are defined based on the contextual relationship between the premise and the hypothesis: entailment, where the premise supports and logically confirms

Corresponding author
İskender Ülgen Oğul,
iskenderogul@iyte.edu.tr

the hypothesis; contradiction, where the premise refutes or negates the hypothesis; and neutrality, where the premise and hypothesis are unrelated or lack a definitive logical connection.

The success of NLI models depends heavily on the availability of large, annotated datasets. While high-resource languages like English benefit from multiple datasets, such as SNLI (Stanford Natural Language Inference) (*Bowman et al., 2015*), MultiNLI (Multi-Genre Natural Language Inference) (*Williams, Nangia & Bowman, 2017*), and MedNLI (*Romanov & Shivade, 2018*), the creation of similar resources for low-resource languages has been significantly constrained by the time and the cost required for data collection and manual annotation. Despite these constraints, a promising alternative to creating datasets from scratch is the machine translation of existing datasets from high-resource languages into low-resource ones.

Although dataset creation through translation is a viable solution, this approach presents unique challenges in specialized domains like medical context. These challenges include the complexity of medical terminology, abbreviation disambiguation, contextual nuances, and metric conversions, which, combined with the domain's highly specialized vocabulary, add further complexity to NLI tasks. Despite these challenges, the growing integration of AI and NLP technologies has increased the demand for high-quality medical datasets. While datasets like the MedNLI (*Romanov & Shivade, 2018*) serve as valuable resources for English language medical NLI development, comparable resources remain scarce, leaving a significant gap for low-resource languages.

In this article, we address the critical gap in the availability of medical NLI datasets for the Turkish language by introducing TurkMedNLI, the first Turkish medical NLI dataset. Following the successful approaches demonstrated by recent studies such as RuMedNLI (*Blinov et al., 2022a*) and ViMedNLI (*Phan et al., 2022*), which have shown the viability of dataset creation through translation from high-resource languages, we propose a novel pipeline that leverages large language models for dataset translation and refinement. Our methodology begins with the fine-tuning of the Llama-3.1 (*Dubey et al., 2024*), a state-of-the-art large language model, using the Medical Abbreviation Disambiguation Dataset for Natural Language Understanding (MeDAL) (*Wen, Lù & Reddy, 2020*). This fine-tuning process ensures accurate identification and expansion of medical abbreviations within sentence pairs, addressing a fundamental challenge in medical text translation. Subsequently, we employ Facebook's No Language Left Behind (NLLB) (*Costa-jussà et al., 2022*) translation model to translate the refined sentences from English to Turkish. The quality of these translations is validated through a two-fold evaluation process: first, the machine evaluation using gold labels from the MedNLI dataset, and secondly, manual evaluation by medical experts. The machine evaluation involves fine-tuning pre-trained language models BERT (*Devlin et al., 2019*), BERTurk (*Schweter, 2020*), BioBERT (*Lee et al., 2019*), and BioBERTurk (*Türkmen et al., 2023*), while domain expert physicians conduct manual evaluation on selected subsets of the translated dataset. This study makes the following contributions:

- We introduce TurkMedNLI, the first comprehensive Turkish Medical Natural Language Inference dataset, addressing a critical resource gap in Turkish biomedical NLP research and enabling the development of domain-specific language understanding models.
- We propose a novel end-to-end pipeline for translating domain-specific datasets into low-resource languages, incorporating medical abbreviation disambiguation, metric correction and context-aware translation mechanisms. This methodology effectively addresses key challenges in medical text translation, including terminology standardization and contextual preservation.
- We demonstrate the efficient use of large language models for dataset refinement and normalization in the medical domain. Our experimental results demonstrate the potential of large language models in medical dataset information extraction and refinement, suggesting improvements over direct translation approaches while providing insights for future cross-lingual dataset creation.

The structure of this article is as follows: The Introduction outlines the significance of NLI research in the NLP field, explores the challenges in developing NLI datasets for low-resource languages, and introduces the TurkMedNLI dataset. The Related Work section reviews prior research on NLI, providing the contextual foundation for this study. The Technical Background section provides an overview of the datasets and models employed, including Llama-3.1 and the No Language Left Behind (NLLB) translation model. The Research Methodology details the process of translating the MedNLI dataset into Turkish, emphasizing the steps involved in refinement and quality control. The Evaluation section presents both machine-based and human evaluations of the translations to assess their quality and reliability. The Results section highlights the improvements and quality achieved with the TurkMedNLI dataset. Finally, the Discussions and Conclusion summarize the key contributions of the study, discusses potential implications, and suggests directions for future research.

## RELATED WORK

NLI is a foundational task in NLP, where the goal is to determine the contextual relationship between a premise and a hypothesis. To support research in this area, datasets like the Stanford Natural Language Inference (SNLI) dataset (*Bowman et al., 2015*) and the Multi-Genre Natural Language Inference (MultiNLI) dataset (*Williams, Nangia & Bowman, 2017*) were created. These datasets have become benchmarks for open-domain NLI task. The construction of large annotated datasets through crowdsourcing is often conducted in English, as it is considered ideal for achieving high-quality results. While effective, this approach presents significant challenges in terms of both cost and time investment. Recent advancements in natural language processing have enabled a practical solution to dataset creation, adapting existing high-quality datasets into low-resource languages through machine translation. A notable implementation of this approach is demonstrated in the study (*Budur et al., 2020*). The research focused on adapting two established datasets, SNLI (*Bowman et al., 2015*) and MultiNLI (*Williams, Nangia & Bowman, 2017*) from English to Turkish. Their approach addressed the challenge of

resource constraints by offering a cost-effective and efficient alternative to creating annotated datasets from scratch. Rather than replicating the labor-intensive process of creating SNLI and MultiNLI, they employed Amazon Translate, a commercial neural machine translation service, to translate these datasets. The translation process was completed in approximately five days at a cost of around $2,000, representing a significant reduction compared to manual dataset creation. To ensure the quality and reliability of the translated datasets, a comprehensive two-step verification process was implemented. In the first step, a team of bilingual Turkish-English speakers conducted validation on the subset of the translated texts, examining both translation accuracy and contextual relation to NLI labels. The second phase employed machine verification through the BERTurk model (*Schweter, 2020*), which achieved notable performance metrics: 85.84% accuracy for SNLI translation, 75.16% and 75.60% for MNLI matched and mismatched sets, respectively. This two-step verification approach demonstrated the effectiveness of the machine translation system in maintaining contextual integrity while producing high-quality translations.

While creating datasets through translation successfully adapts general-domain NLI datasets like SNLI and MultiNLI, specialized domains such as healthcare require more tailored approaches. The medical domain, in particular, presents unique challenges, including the need for precise interpretation of complex terminologies, specialized vocabulary, and nuanced contextual relationships. Recognizing this need, researchers introduced MedNLI, the first large-scale, physician-annotated clinical language inference dataset. MedNLI specifically addresses the limitations of general-domain datasets such as SNLI and MultiNLI, which lack the complexity and contextual relationships needed for medical applications. MedNLI (https://physionet.org/content/mednli/1.0.0/) established a foundation for domain-specific NLI research by focusing on issues like medical terminology, abbreviations, and expert interpretation. The dataset is available on PhysioNET (*Goldberger et al., 2000*), with CITI Program certification, and requires credentialed approval access. The creation of MedNLI followed a systematic three-phase methodology to ensure quality. The development process follows premise extraction, expert annotation, and quality verification, as detailed below.

**Premise sampling:** In the initial phase of premise sampling, clinical text were extracted from the MIMIC-III database (*Johnson et al., 2016*), focusing on the Past Medical History sections of deidentified clinical records. These texts were then processed using a biomedical sentence splitter to identify suitable premise candidates, ensuring the selection of clinically relevant content.

**Annotation process:** Once the premises were extracted, clinicians generated hypotheses for each premise that were contextually entailed, neutral, or contradictory. This annotation process followed the standard methodology established by the SNLI dataset, which has become a widely adopted framework for NLI dataset creation.

**Quality assurance:** Quality assurance was implemented through a detailed two-step verification process. The human verification involved clinical experts examining the hypothesis statements for both annotation consistency and medical accuracy. Physicians effectively identified and removed examples containing artifacts or insufficient contextual

information. The reliability of these annotations was quantitatively assessed using Cohen's kappa coefficient. To further assess the dataset's quality, machine verification was conducted using models such as Bag of Words (*Zhang, Jin & Zhou, 2010*), InferSent (*Conneau et al., 2017*), and ESIM (*Chen et al., 2017*). Machine evaluation steps achieved accuracy of 51.9%, 71.9%, and 76.0%, respectively. These results validated the reliability of the annotations and the overall quality of the dataset.

Following the advancements in adapting NLI datasets to new languages, researchers have explored the feasibility of extending domain-specific datasets like MedNLI to low-resource languages. While the Turkish adaptations of SNLI and MultiNLI showed the potential of translation-based methods for general-purpose datasets, the RuMedNLI study (*Blinov et al., 2022a*) demonstrated a similar approach tailored to the medical domain. Addressing the scarcity of Russian medical text resources, particularly in natural language inference, the RuMedNLI project translated the MedNLI dataset into Russian, creating the first Russian medical NLI dataset. This work has become part of RuMedBench (*Blinov et al., 2022b*), a benchmark for Russian medical language understanding. The RuMedNLI creation process employed multiple processes to ensure consistency and correctness. MedNLI was translated independently using two automatic translation services, Google Translate and DeepL, and then manually reviewed for quality. The review process included Russian-English bilingual contributors, consulted by a medical team with extensive experience in Russian clinics and the medical domain. The review procedure included picking the best translation and making the required changes to correct the issues, as shown below.

- **Translation inaccuracies:** Errors from incorrect interpretation of medical contexts when using direct translation processes.
- **Abbreviation misinterpretations:** Instances where medical abbreviations or terms were mistranslated, leading to loss of intended meaning.
- **Unit conversion errors:** Errors in translating measurements, such as converting "feet" to "meters," resulting in inconsistencies.

To ensure the quality of the translated dataset, the researchers developed a comprehensive evaluation approach using both human and machine evaluation methods. Human evaluation analyzed a detailed review of a subset of the dataset by bilingual experts, while machine assessment employed NLP models such as ESIM (*Chen et al., 2017*) to evaluate the contextual and semantic alignment of the translated pairs. Notably, only 32.3% of the automatically translated texts were acceptable without changes, while the rest of the sentences required revisions. This highlights the inherent challenges in accurately translating medical terminology and context across languages. RuMedNLI has utilized various architectures, including BERT-based models. For instance, the RuBERT model, a BERT adaptation for the Russian language, achieved an accuracy of 77.64% on the RuMedNLI dataset. Another model, RuPoolBERT, attained an accuracy of 77.29%. The RuMedNLI was the first research to successfully transfer a medical context NLI dataset into another language.

Following a similar approach to RuMedNLI, researchers developed ViMedNLI (*Phan et al., 2022*) to address the scarcity of Vietnamese medical language resources. Their work similarly adapted the MedNLI dataset through a structured translation and validation pipeline. The process began with the application of state-of-the-art English-Vietnamese neural machine translation models, followed by a systematic evaluation framework. The evaluation process incorporated both human expertise and machine learning methods, which is a standard approach. Domain experts, specifically Vietnamese pre-medical students, refined the translations following comprehensive guidelines that emphasized precise medical terminology and contextual accuracy. This process also addressed similar issues to those encountered in RuMedNLI, including direct translation errors, medical abbreviations, measurement standardization, and cultural adaptations necessary for the Vietnamese medical context.

To further evaluate the ViMedNLI dataset, machine verification was conducted using advanced language models tailored to Vietnamese biomedical text, including ViHealthBERT (*Nguyen et al., 2022*) and ViPubmedT5 (*Phan et al., 2022*). The ViPubmedT5 model, a T5-style encoder-decoder Transformer pretrained on synthetic biomedical data, achieved state-of-the-art accuracy of 81.65%, while ViHealthBERT attained 79.04% accuracy on the translated dataset. These results highlight the effectiveness of domain-specific pretrained language models, particularly ViPubmedT5, for natural language inference in Vietnamese medical texts. The results demonstrate that ViMedNLI successfully enriches Vietnamese biomedical NLP research by providing a robust and high-quality resource for further studies.

Beyond NLI, recent advancements in multilingual medical NLP emphasize the development of language resources for non-English languages. Recent surveys outline the creation of datasets and models tailored to various languages and medical domains (*Shaitarova et al., 2023*). The democratization of large language models, combined with their increasing accessibility, has opened new possibilities (*Grouin & Grabar, 2023*). Furthermore, specialized models such as ClinicalGPT (*Wang et al., 2023*), designed and optimized for clinical scenarios, showed how large scale fine-tuning with domain-specific medical data can address challenges like reasoning and dialogue generation in healthcare settings. These efforts collectively highlight the transformative potential of NLP in addressing diverse challenges within medical research and practice.

Our study builds on previous research in the field of NLI, leveraging the advances made by earlier studies. Creating NLI datasets from scratch requires significant time, effort, and resource. However, adapting these datasets into another language using translation methods presents a feasible solution, as demonstrated by SNLI-tr and MultiNLI-tr. Although translation adaptation has been effective, domain-specific NLI datasets face unique challenges, as addressed by RuMedNLI and ViMedNLI. Following prior research, our study introduces a novel pipeline that uses large language models not only for translation but also for systematically resolving previously identified issues in adapting datasets for low-resource settings.

# TECHNICAL BACKGROUND

In this section, we start by giving an overview of the datasets used in our study, such as SNLI, MultiNLI, and MeDAL, which form the core resources for our analysis. We then explain the technical background of the language models used, focusing on their key principles and how they align with the goals of our research.

## NLI datasets

The lack of a large annotated dataset affected the development of models capable of precisely understanding semantic relationships between text pairs. The Stanford Natural Language Inference SNLI *corpus* was developed to address these issues and provided a gold standard. SNLI is composed of 570,152 sentence pairs that have three classification labels. These labels are identified as entailment where sentences are complementary to each other, contradiction where sentences reject each other, or a state where sentence pairs are neutral to each other. An example of the dataset is given in Table 1 providing standard NLI dataset structure.

The sentence pairs in the SNLI dataset were generated through a detailed crowdsourcing process using Amazon Mechanical Turk (*Buhrmester, Talaifar & Gosling, 2018*). Contributors were provided with image captions and instructed to create sentences that were categorized as entailment, neutral, or contradiction relative to the original sentence. With this approach original sentences are based on real-world observations and enriched the dataset with authentic and diverse examples. To ensure label accuracy and annotator agreement, each sentence pair's initial label was reviewed by four additional annotators. The final gold label was determined by majority vote, reflecting the most reliable consensus among the five annotations. This validation process resulted in a 98% consensus rate with high label reliability. The creation of SNLI not only provided a critical resource for semantic comprehension difficulties, but it also set new standards for dataset size and annotation quality in the field of NLI.

Although SNLI is a profound source for the NLI task, the dataset primarily consists of a single text source, which lacks a broad spectrum of genre information. The MultiNLI (*Williams, Nangia & Bowman, 2017*) aimed to address these limitations. To propose a solution to these issues, the MNLI *corpus* expands on context diversity and complexity by including a wider range of textual genres. The goal of the multi genre dataset is to better represent the wide range of linguistic backgrounds that reflect real life. This will help build strong bases for NLI models. MultiNLI includes sentences from a wide range of written and spoken English genres, such as face-to-face conversations, government reports, letters, travel guides, fiction, magazine articles and various texts from the Open American National Corpus. The diversity of the data sources creates a broad spectrum of language occurrences, increasing the complexity and difficulty of the inference tasks compared to SNLI. The variable data source structure of MultiNLI indicates that it is more challenging than SNLI, highlighting its potential to improve the field of the NLI research area.

While general purpose NLI datasets such as SNLI and MultiNLI have laid the foundation for advancements in natural language inference, domain specific datasets like MedNLI are essential for addressing the unique complexities of the medical field. It is

**Table 1 SNLI *corpus* sample (source: *Bowman et al., 2015*).**

| Premise | Label | Hypothesis |
|---|---|---|
| A black race car starts up in front of a crowd of people. | *Contradiction* | A man is driving down a lonely road. |
| An older and younger man smiling. | *Neutral* | Two men are smiling and laughing at the cats playing on the floor. |
| A soccer game with multiple males playing. | *Neutral* | A happy woman in a fairy costume holds an umbrella. |

**Table 2 Comparison of dataset characteristics for SNLI, MultiNLI, and MedNLI.**

| Characteristic | SNLI | MultiNLI | MedNLI |
|---|---|---|---|
| **Dataset size** | | | |
| Training pairs | 550,152 | 392,702 | 11,232 |
| Development pairs | 10,000 | 10,000 (Matched) | 1,395 |
| Test pairs | 10,000 | 10,000 (Mismatched) | 1,422 |
| **Average sentence length (Tokens)** | | | |
| Premise | 14.1 | 22.3 | 20.0 |
| Hypothesis | 8.3 | 10.6 | 5.8 |
| **Maximum sentence length (Tokens)** | | | |
| Premise | 82 | 401 | 202 |
| Hypothesis | 62 | 70 | 20 |

tailored for clinical applications and provides general linguistic resources and the specialized needs of medical text processing. MedNLI stands out from SNLI and MultiNLI in terms of size and structure, as shown in Table 2, highlighting the challenges of developing annotated datasets in the medical domain. Unlike general domain NLI datasets, MedNLI incorporates medical terminology and abbreviations, which often have multiple meanings depending on their position and the context, creating unique challenges for machine learning models and cross domain adaptation. Having more than one meaning also makes dictionary based solutions impractical. To address this issue and propose a solution, we also utilized the MeDAL (*Wen, Lù & Reddy, 2020*) dataset, which is derived from PubMed abstracts and focuses on medical abbreviation disambiguation. MeDAL provides high-quality annotations to clarify the meaning of ambiguous terms that enhance the capacity of a model and interpret complex representations.

The MeDAL dataset was derived from PubMed abstracts, using a systematic approach to ensure both scale and quality. This process began with the collection of articles from PubMed. Later, to generate accurate labels without manual annotation, the team employed the reverse substitution technique, replacing full terms with their corresponding abbreviations using the ADAM database (*Zhou, Torvik & Smalheiser, 2006*). To manage the dataset's scale while maintaining balanced representation, a strategic sampling method was applied to create computationally feasible training, validation, and test sets.

Table 3 illustrates the structure of the MeDAL dataset and three key fields. The *text* field contains sentences or paragraphs with abbreviated terms. The *location* indicates the word index where the abbreviation occurs. Finally, the *label* field provides the expanded version

**Table 3  MeDAL dataset example (source: _Wen, Lù & Reddy, 2020_).**

| Text | Location | Label |
|---|---|---|
| velvet antlers vas are commonly used in traditional Chinese medicine and invigorant… | [63] | ["transverse aortic constriction"] |
| the clinical features of our cases demonstrated some of the already known… | [85] | ["hodgkins lymphoma"] |
| ceftobiprole bpr is an investigational cephalosporin with activity against… | [90] | ["methicillininsusceptible"] |

of the abbreviation within the given context. While the training, development, and test splits offer a single abbreviation per example, the complete raw dataset includes examples with multiple abbreviations. This detailed structure of the dataset preserves multiple abbreviations per entry, offering a more comprehensive resource for abbreviation disambiguation.

## Language models

The language models used in our study are primarily based on transformer networks, which are designed around the self-attention (_Bahdanau, Cho & Bengio, 2014_) mechanism. Transformer networks differed greatly from previous sequential algorithms such as LSTM (_Hochreiter & Schmidhuber, 1997_), which rely on recurrent neural networks. Unlike sequential processing, transformer architecture eliminates recurrent structures, which results in improved parallelization and reduced computational cost. Transformers' structure consists of stacked encoders and decoders, each incorporating a multi-head self-attention mechanism and a fully connected feed-forward network (_Vaswani et al., 2017_). The multi-head attention mechanism enables the model to capture long-range dependencies, leading to a comprehensive understanding of context for natural language processing tasks. The success of the transformer architecture was quickly adopted by the community and paved the way for pre-trained NLP models such as BERT (_Devlin et al., 2019_), GPT (_Brown et al., 2020_), and large language models such as the Llama model family (_Dubey et al., 2024_).

**Meta Llama-3.1:** Meta has made great contributions to the open-source LLM community with the Llama model family, which presented a significant advancement over its predecessor. Llama-3.1 preserves the foundational architectural design of Llama-2 while integrating key enhancements that expand its capacity and capability. Their new approach introduces an expanded vocabulary of up to 128,000 tokens, which enables the model to handle more extensive contexts and improve multilingual capabilities. This improvement is complemented by the sophisticated Grouped Query Attention mechanism (_Ainslie et al., 2023_), which allows for efficient handling of sequences with up to 128k tokens in context length.

The model's quality is also determined by the quality and content of the training datasets (_Budach et al., 2022_). To achieve quality, Meta carefully assembled a pre-training dataset of 15 trillion tokens, a seven-fold increase in _corpus_ size compared to Llama-2. This dataset not only enhanced the linguistic variety by including texts from 30 languages but also prioritized the high quality of multilingual assistance. In order to guarantee the

accuracy and dependability of the dataset, Meta incorporated an extensive range of data filtering techniques, such as heuristic filters, filters for NSFW material, semantic deduplication algorithms, and powerful text classifiers. The classification process has been improved by utilizing the existing capabilities of Llama-2 to generate and categorize high-quality textual input.

In addition, Meta has improved its post-training procedures to enhance Llama-3.1's performance in interactive applications, particularly in chat-based contexts. The model goes through a precise procedure for adjusting its instructions and alignment, which involves Supervised Fine Tuning (SFT) (*Ouyang et al., 2022*), Rejection Sampling, Proximal Policy Optimization (PPO) (*Schulman et al., 2017*), and Direct Preference Optimization (DPO) (*Rafailov et al., 2023*). These methods have played a crucial role in improving the model's performance in reasoning and responding (*Dubey et al., 2024*).

The choice of the Llama-3.1 8B model over alternative LLMs was driven by its exceptional reasoning capabilities. The model achieved competitive scores in tasks requiring advanced reasoning and comprehension, such as 69.4% accuracy on MMLU and 73.0% on MMLU (CoT, Chain-of-Thought). Moreover, its extensive and diverse pre-training dataset, coupled with sophisticated post-training fine-tuning methodologies, ensures consistently high quality performance across a wide range of tasks. These metrics and design choices collectively highlight Llama-3.1's capacity for precise and reliable language understanding, making it an optimal choice for tasks requiring sophisticated contextual reasoning.

**Facebook NLLB (No Language Left Behind):** Machine translation has traditionally focused on high-resource languages due to the availability of large training datasets. In contrast, low-resource languages, which make up the majority of global languages, have often been left behind due to data scarcity and challenges in data preparation. The NLLB (*Costa-jussà et al., 2022*) model can translate between 200 languages, supporting both low and high resource languages. The project aimed to improve linguistic equity more effectively than traditional methods by employing advanced data curation techniques and an encoder-decoder transformer architecture. This approach enabled the creation of a sequence-to-sequence translation model designed to handle both low-resource and high-resource languages. To achieve high quality performance, the NLLB model relied on a carefully curated dataset. Thus, the Flores-200 dataset (*Costa-jussà et al., 2022*) was developed as an extended and enhanced version of the earlier Flores-101 dataset (*Goyal et al., 2021*). Flores-200 not only significantly increases the number of languages covered but also ensures that translations are culturally appropriate and accurate.

Evaluation of the NLLB uses both automatic and human-assisted metrics to assess translation quality, focusing on safety and precision. Conventional metrics, such as BLEU (*Papineni et al., 2002*), are typically inappropriate for low-resource languages because of the absence of large parallel datasets and the challenges associated with tokenization. In order to address these limits, the NLLB project utilizes ChrF++ (*Popovic, 2015*) and spBLEU (*Costa-jussà et al., 2022*) scores. To provide a more comprehensive quality assessment, the ChrF++ score considers character-level n-grams, as well as word uni-grams and bi-grams. Furthermore, spBLEU, which aims to reduce tokenization bias, is

used in conjunction with the SentencePiece (*Kudo & Richardson, 2018*) model. In addition to machine evaluation, the NLLB team used human assessment based on a variety of criteria, such as fluency, adequacy, and translation accuracy. This comprehensive evaluation provided a holistic assessment of machine translation performance and presented a state-of-the-art language translation model. Considering the advancement that the model brings and its ability to support low-resource languages with high quality translation, the NLLB model offered a balanced and reliable solution for our translation step.

The BERT model fundamentally changed natural language processing by allowing for the contextual representation of a word based on the words that come before and after it. Following BERT's success, researchers created various specialized adaptations for specific topics and languages. For instance, BioBERT trained on massive biomedical datasets to process biomedical texts, significantly enhancing its capacity to interpret and analyze complex medical language. Similarly, BERTurk was built to handle Turkish's distinct language qualities, including its morphological and syntactic aspects. BioBERTurk adapts the pre-training methodology of BioBERT with BERTurk's linguistic optimizations to improve the efficiency and accuracy of biomedical text analysis in Turkish.

**BERT:** The development of BERT presented a significant advancement in the NLP world. It introduces a revolutionary model that effectively captures a document's contextual information. The model achieves the ability to express the context of a word by considering the surrounding words, resulting in a more comprehensive understanding of language syntax and semantics. This model is trained employing the masked language model technique, which involves masking individual words inside a sentence and predicting them based on their context. BERT was pre-trained on a large amount of text material, such as the English Wikipedia and BookCorpus, allowing it to gain comprehensive and contextual language knowledge. As a result, BERT showed exceptional performance on a variety of NLP tasks, including question answering, sentiment analysis, and language inference. Its adaptability had allowed other BERT-based models to be tailored to individual languages and specialized topics.

**BioBERT:** BioBERT is a domain-specific adaptation of the BERT model that improves the performance of biomedical natural language processing tasks. BioBERT builds on the pre-trained BERT model, expanding its training to include large biomedical corpora such as PubMed abstracts and PMC full-text articles. These sources include a vast amount of biological knowledge, with PubMed abstracts comprising around 4.5 billion words and PMC full-text articles adding approximately 13.5 billion words. BioBERT has been tested on a variety of common biomedical text mining applications, including named entity recognition, relation extraction, and question answering. In these tests, BioBERT beat baseline models, establishing new state-of-the-art benchmarks and demonstrating its ability to comprehend and analyze complicated biomedical literature.

**BERTurk:** The BERTurk model is a modified version of the original BERT model that addresses the Turkish language's unique linguistic characteristics. Using the BERT model architecture, BERTurk is trained on a large Turkish *corpus*. Thus, it allows the model to adapt to the unique syntactic and semantic complexities of Turkish. The training data

includes a filtered and sentence-segmented version of the Turkish OSCAR (*Abadji et al., 2022*) *corpus*, as well as a Wikipedia dump, which provide a solid foundation for understanding and processing Turkish text.

**BioBERTurk:** BioBERTurk is a specialized language model that improves the analysis of biomedical text in Turkish, a language known for its complex morphology. The model was trained on a large dataset, which included Turkish medical publications and radiology theses obtained from Dergipark and the Turkish Council of Higher Education's database. BioBERTurk aims to improve comprehension of domain specific language, which is critical for resource constrained NLP tasks. The model's performance was evaluated through a classification task on radiology reports, where it demonstrated better results compared to generic language models and baseline approaches. These findings highlight BioBERTurk's ability to effectively capture the context of Turkish biomedical language and provides essential solution for biomedical research and practical applications.

While recent advancements in transformer-based models have introduced more complex architectures, the decision to use BERT and its domain-specific adaptations, such as BioBERT and BERTurk, was driven by several critical factors. First, the availability of pre-trained models specifically tailored to Turkish language and biomedical text is limited. As of now, to our knowledge, BERT-based models, including BERTurk and BioBERT, remain the most robust and accessible options for research involving Turkish datasets. These models allow us to evaluate each stage of our research using a unified model architecture, ensuring consistency and comparability throughout the process. Additionally, their proven performance across diverse NLP tasks, coupled with the flexibility to adapt them to specific domains, provides a solid foundation for addressing the challenges in Turkish biomedical NLP. While exploring more advanced models may be a consideration for future work, the use of BERT-based models was a practical and effective choice given the scope and objectives of this study.

## RESEARCH METHODOLOGY

This study employed large language models (LLMs) in a systematic sequence to translate the MedNLI dataset into Turkish, establishing a comprehensive and robust pipeline. As illustrated in Fig. 1, the initial phase involved acquiring appropriate datasets for translation task. The MedNLI dataset contains domain-specific abbreviations and terminology, which require careful handling during information extraction and translation. To address this challenge, the Medical Abbreviation Dataset (MeDAL) was utilized to fine-tune Llama-3.1.

The selection of an appropriate language model capable of handling complex tasks was a crucial step. Our research decided to use the Llama-3.1 model for its advanced reasoning capabilities, specifically fine-tuning it with the MeDAL dataset to extract medical abbreviations and their expanded forms from clinical sentences. This approach leveraged the model's sophisticated language understanding abilities, as discussed in our previous analysis of the Llama architecture. Following the abbreviation extraction and sentence refinement phase, we employed the NLLB-200 model for Turkish translation. To evaluate our entire pipeline, the BERT model and its variants, BioBERT, BERTurk, and BioBERTurk, were employed to assess the viability of the refined and translated sentence

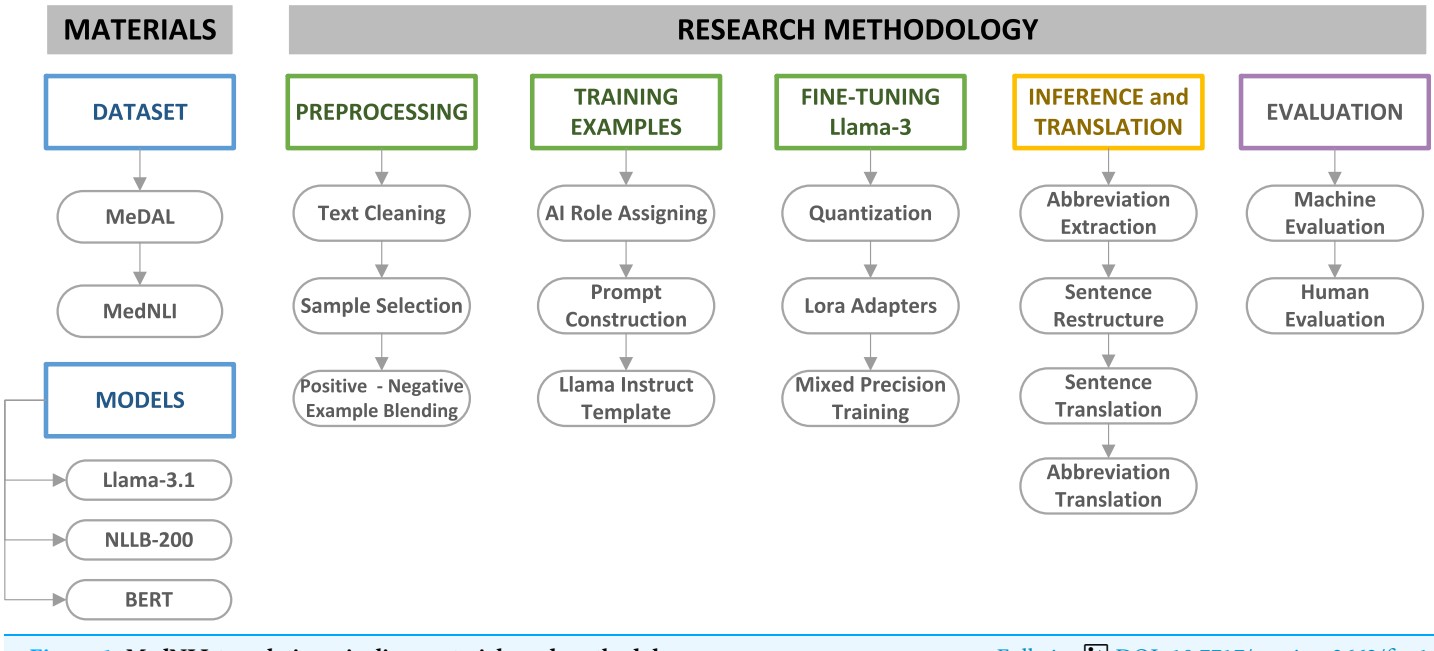

**Figure 1 MedNLI translation pipeline materials and methodology.**

results. Domain-specific datasets, especially in the medical domain, require careful assessment. In addition to machine evaluation, human evaluation was conducted on a statistically significant subset of the dataset to validate the overall translation quality and feature extraction accuracy.

## Preprocessing

The MedNLI dataset is created using the MIMIC-III (*Johnson et al., 2016*) *corpus*, which is compiled from real patient records. Using patient records to create an open source dataset requires extensive de-identification steps to prevent access to information such as real patient names, doctor names, or hospital names. The de-identification steps use tags like [**Person Name**] to conceal personal information in the sentences. These tags do not hold any meaning for NLI tasks and can affect the quality of models developed using this dataset. To maintain the optimal quality of the models, these tags have been cleaned from the MedNLI using an advanced regex text matching system. This preprocessing step ensures that all tags within [** **] were removed from the dataset. Thus, the MedNLI dataset became suitable for feature extraction and translation.

Before translating the MedNLI, there are several issues that need to be addressed, such as abbreviations, measurement differences like lengths from feet to meters, and weights from pounds to kilograms. In our work, we approached abbreviations as an information extraction opportunity since they can convey meaning and influence the alignment of two sentences. Therefore, we proposed a solution that utilizes LLMs to extract abbreviations and their expanded forms while taking context into account. Given the significant computational resources required for fine-tuning large language models, careful curation

of training examples from MeDAL was essential to ensure both quality and efficiency before fine-tuning Llama-3.1. Although the MedNLI dataset consists of longer sentences compared to the SNLI dataset, it is worth noting that the average sentence length in MedNLI is 20 tokens, with a maximum length of 202 tokens, as represented in Table 2. This implies that abbreviation extraction will be the task of retrieving information from short sentences. Given this, we employed fundamental analytical steps to create meaningful training examples.

To ensure both computational efficiency and training effectiveness, we implemented a systematic approach to select appropriate training examples from the MeDAL dataset. Our methodology began with establishing three fundamental metrics for each example: word count, abbreviation count, and abbreviation density. The abbreviation density measurement $AD$ is defined by:

$$AD = \frac{A}{W} \tag{1}$$

where $AD$ represents the abbreviation density measurement bounded by $[0, 1]$, $A$ denotes the total number of abbreviations in the example $A \in \mathbb{Z}^+$, and $W$ represents the total word count $W \in \mathbb{Z}^+$. Abbreviation density was calculated as the ratio of abbreviation count to word count. We then employed box plot analysis to identify and remove outliers in both abbreviation density and word count distributions. Examples that exceeded the $IQR$ range for both metrics are pruned from the selection process. Lastly, we constructed boundaries for word counts and abbreviation density, examples longer than 40 words and with abbreviation density higher than 0.5 are pruned.

This simple and effective filtering approach yielded approximately 800,000 training examples. The decision to maintain an abbreviation density threshold of 0.5 was strategically chosen to prevent overwhelming the model with abbreviation-heavy examples while maintaining a balanced representation of medical text patterns. Additionally, our focus on shorter sentences aligns with both computational efficiency requirements and the characteristics of our target dataset. MedNLI exhibits relatively short average token lengths. In a way we ensure that our model learns to extract information from contexts similar to the one it will encounter later.

One of the key considerations when fine-tuning language models is to employ a versatile training approach that includes both positive and negative examples. This enables the model to learn effectively from a diverse range of scenarios. Training the model exclusively with positive examples would condition it to assume every input sentence contains an abbreviation, leading to incorrect responses. To address this, we split the samples into 60% positive and 40% negative examples. The divide ratio is calculated heuristically, since the RuMedNLI study corrected 10,110 examples, indicating that there are numerous examples of abbreviations and metric differences. This approach ensures that the model can learn to correctly resolve any detected abbreviations, and if there are none, remain the sentence unchanged.

## Training examples

After pretraining, LLMs, like Llama-3.1, undergo a process known as instruction tuning. This process enables the raw model to interact with users in a structured conversational format. Instruction tuning involves fine-tuning the model on a large optimization dataset and defining specific conversational roles. This role-based fine-tuning allows the model to effectively learn and understand the structure of the conversation with users.

Llama-3.1 uses three distinct roles within its conversational framework: *system*, *user*, and *assistant*. The system role issues prompts to the model, defining the task for the model's responses. The user role represents the input from the user. The assistant role constitutes the model's response, generated based on the given commands and user input.

The system role expects a prompt that provides clear instructions for the model. Creating a single prompt for all examples could potentially overfit the model, leading it to memorize the task instead of understanding it. To alleviate this issue, we have created 40 different prompts for positive examples and 30 different prompts for negative examples. Positive prompts instruct the model to find abbreviations in the text, whereas negative prompts instruct the model to leave the sentences as they are.

To ensure uniform training representation, we implemented a cyclic distribution method where 40 distinct prompts were assigned sequentially to examples in a fixed order. This systematic approach prevented prompt frequency imbalance by restarting the sequence from the first prompt after reaching the 40th prompt, continuing until all examples received a prompt assignment. This balanced approach was critical for preventing any one of the prompts from skewing the learning process. Following the same logic, 30 negative prompts were iterated through non-abbreviated examples. We have created a chat template for a positive example using MeDAL, as demonstrated in Table 4.

During the later stages of the fine-tuning phase, we realized that assigning a specific role to the AI at the beginning of the prompt resulted in a reduction of 0.10 in the training loss. For instance, we assigned the role, "*As a medical examiner, your task is to analyze and clarify medical texts with precision and accuracy. Ensure your responses adhere to this guideline.*" This simple instruct suggests that LLMs perform more effectively when fine-tuned for a specific task with a clearly defined role.

## Fine tuning Llama-3.1

Due to the nature of the LLMs and their large parameter space, it is not quite possible to fit an 8 billion parameter model into a single GPU. To mitigate this, optimization techniques such as quantization have gained importance in recent years. Quantization reduces the model's weights from $32 - bit$ precision to as low as $4 - bit$ precision with the aim of minimal loss of the model quality. This process saves a significant amount of GPU memory and enables the training phase to process longer context sizes with large batches. For example, quantization of $32 - bit$ floating point information to $int8$ using the $c$ quantization constant with a range $[-127, 127]$ is calculated as:

**Table 4 Llama-3.1 fine tuning instruction format.**

```
{"role": "system",

"content": "Detect all abbreviations present in the text and produce two lists: one for the abbreviations and another for
  their full expansions."},

{"role": "user",

"content": "the nutcracker esophagus a primary motor disorder is frequently associated with noncardiac CP" },

{"role": "assistant",

"content": "**abbreviations**: [CP] **abbreviations_extended**: [chest pain]"}
```

$$\mathbf{X}^{\text{int8}} = \text{round}\left(\frac{127}{\text{absmax}(\mathbf{X}^{\text{FP32}})} \cdot \mathbf{X}^{\text{FP32}}\right) = \text{round}(c^{\text{FP32}} \cdot \mathbf{X}^{\text{FP32}}) \tag{2}$$

In our research we've employed QLora (Quantized Low Rank Adapters) (*Dettmers et al., 2023*), which leverages NF4 NormalFloat4 quantization to significantly reduce GPU memory requirements. NF4 quantization involves estimating the quantiles of a normal distribution and mapping the values to a $4-bit$ format. This ensures more efficient utilization of quantization bins, particularly for data with a zero-mean normal distribution. As a result, we managed to fit a model such as the Llama-3.1 with 8 billion parameters into the Nvidia RTX 4090 GPU.

Fully fine-tuning large language models presents significant computational challenges, especially with limited GPU power. To address this constraint, we implemented the Low-Rank Adaptation (LoRA) (*Hu et al., 2021*) technique, which offers a more efficient alternative by placing small trainable adapter layers between the model's existing layers.

LoRA modifies the neural network architecture through a systematic decomposition of the weight update process. This approach utilizes two key components: a pre-trained weight matrix $W \in \mathbb{R}^{d \times d}$ that remains frozen during training, and a LoRA weight matrix $\Delta W$ that is decomposed into two low-rank matrices $A$ and $B$. The matrix $A$ is initialized with a normal distribution $N(0, \sigma^2)$, while $B$ is initialized as zero. During fine-tuning, only the matrices $A$ and $B$ are subjected to training, creating a low-rank update to the original weights. This design enables task-specific adaptation while preserving the model's core knowledge.

Combining quantization with low rank adapters, it is now possible for a single GPU to fine tune a model for a specific task. To train efficiently with a minimal memory footprint, we used mixed precision training. Mixed precision training uses Float16 or BFloat16 for newer systems. In our work, we used the parameters demonstrated in Table 5.

The LoRA rank parameter was set to 32, with an alpha scaling factor of 8, while maintaining default values for dropout and bias parameters. To optimize memory utilization during training, we employed gradient checkpointing through the Unsloth library, which strategically stores selective activations in the forward pass while recalculating others during backpropagation. Additionally, we standardized our training

**Table 5 Parameters used for Lora adapters.**

| Parameter | Value |
| --- | --- |
| r (rank) | 32 |
| lora_alpha | 8 |
| lora_dropout | 0 |
| bias | None |
| use_gradient_checkpointing | Unsloth |
| random_state | 42 |
| max_seq_length | 2,048 |

process with a random seed of 42 and constrained the maximum sequence length to 2,048 tokens.

Table 6 outlines our training parameters for the supervised fine-tuning trainer. To ensure efficient training, we used BFloat16 precision for both training and evaluation. After careful testing of our GPU memory capacity, we set the batch size to 8. Every 100 steps, we evaluate and checkpoint the model. We set our learning rate to 2e−4, with a maximum gradient normalization of 0.3 and a warm-up ratio of 0.03. We used a linear learning rate scheduler for efficiency. As highlighted in the NEFTune (*Jain et al., 2023*) article, introducing noise to the training procedure can positively impact accuracy; thus, we set NEFTune noise alpha to 5. A key aspect that accelerated the training process was using the group by length parameter, which merges training examples up to 2,048 tokens. Since we use short examples up to 40 words, this procedure of packing examples reduced the training time nearly tenfold. Lastly, we used an eight-bit Adam optimizer and fine-tuned the Llama-3.1 model. We trained our model with three epochs and approximately 5,000 steps.

## Llama-3.1 inference and MedNLI translation

The MedNLI dataset translation process consists of three carefully orchestrated sequential steps, forming our comprehensive medical inference dataset translation pipeline. At the core of our approach is the novel treatment of medical abbreviations as valuable information rather than obstacles. The pipeline begins with our fine-tuned Llama-3.1 model performing automated abbreviation extraction and expansion. In the first phase, we addressed a key structural characteristic of the MedNLI dataset: each premise appears three times, paired with different hypotheses. Recognizing this pattern, we optimized our processing by grouping identical premises together. This strategic grouping served two crucial purposes: it significantly reduced computational overhead and ensured consistent abbreviation handling across related sentence pairs. Thus, our fine-tuned Llama-3.1 model processed each premise group systematically and identified and expanded all medical abbreviations. Following the premise processing, we applied the same methodical approach to the hypothesis statements. This process ensured that both components of each premise-hypothesis pair received consistent abbreviation processing, maintaining coherence throughout the dataset.

**Table 6 Supervised fine tuning training parameters.**

| Parameter | Value |
| --- | --- |
| per_device_train_batch_size | 8 |
| gradient_checkpointing | True |
| num_train_epochs | 3 |
| optim | Adamw_8bit |
| bf16 | True |
| bf16_full_eval | True |
| eval_steps | 100 |
| logging_steps | 20 |
| save_steps | 100 |
| per_device_eval_batch_size | 4 |
| learning_rate | 2e−4 |
| max_grad_norm | 0.3 |
| warmup_ratio | 0.03 |
| lr_scheduler_type | Linear |
| seed | 42 |
| weight_decay | 0.01 |
| neftune_noise_alpha | 5 |
| group_by_length | True |

In the second stage of our pipeline, we leveraged Llama-3.1's inherent capabilities rather than using its fine-tuned version. The model demonstrated robust performance in handling both mathematical computations and context-aware tasks through advanced prompting techniques and one-shot learning approaches. To address the previously identified challenges in medical text processing, we developed a comprehensive prompt structure that systematically handles abbreviations, metric conversions, and contextual nuances. This prompt includes previously extracted abbreviations, detailed instructions, and a carefully crafted example to guide the model's responses. Table 7 illustrates this prompt structure and demonstrates how it effectively guides the model in processing medical text. As a result, we achieve restructured and normalized medical sentences where abbreviations are expanded and metric conversions corrected for the target language.

In addition to the positive examples where abbreviations have been identified, we also addressed negative examples where no abbreviations are present. For these cases, we constructed a similar prompt that focuses solely on metric conversions. If no changes are required, we prompt the model to return the exact same sentence without any modifications.

In the final phase of our pipeline, we implemented Facebook's NLLB model to perform English to Turkish translation of the processed sentence pairs, extended abbreviations, and base sentences. NLLB provides a state-of-the-art translation system with the capability to handle 200 different languages through its language tag architecture. The translation step followed the same design as for the premises, where we grouped sentences to reduce

**Table 7 LLama inference prompt example.**

**Task description:**

As a specialized assistant in medical text refinement, your task is to enhance the readability and precision of the given text without changing its essential content. Perform the following specific tasks carefully:

**Abbreviations expansion:** Identify and expand all medical abbreviations in the text. For example, change 'HTN' to 'hypertension'. To assist you, we have identified some abbreviations in the text along with their extended meanings.

Here are the abbreviations that we could find in the text: {abbreviations}

Here are the longer forms of the abbreviations: {extended_abbreviations}

**Unit conversions:** Convert all relevant units while preserving the original numerical values:

– Temperatures from Fahrenheit to Celsius.

– Lengths from feet to meters.

– Weights from pounds to kilograms.

– Speeds from miles to kilometers.

– AM and PM to 24-h format.

Some sentences may be short and simple, while others may be long and complex.

**Example:**

- **Original:** 'The patient has HTN, weighs 180 lbs, and their temperature was 101 °F.'
- **Refined:** 'The patient has hypertension (HTN), weighs 81.65 kg, and their temperature was 38.33 °C.'

computational demand and increase consistency. After we finished translating the premises, we translated all of the hypotheses. Additionally, we translated the expanded forms of the abbreviations we had previously extracted. This method not only adapted the dataset into Turkish but also laid the groundwork for future natural language processing tasks, such as named entity recognition, by including medical abbreviations, their expanded forms, and their Turkish equivalents.

## Evaluation

The MedNLI translation consisted of multiple steps, and each step provided new aspects and information from the dataset. To be able to assess the quality and reliability, our study employed a two-stage evaluation process.

Our evaluation methodology began with a comprehensive machine learning assessment using multiple BERT-based models. We established a systematic evaluation pipeline consisting of four distinct stages. First, we utilized the standard BERT model to establish baseline performance metrics on the MedNLI dataset. Second, we leveraged BioBERT, a medically fine-tuned variant of BERT, to evaluate the domain-specific aspects of the MedNLI sentence pairs. Third, we assessed our Turkish translations using both BERTurk and BioBERTurk models, which are specifically optimized for Turkish language processing. This multi-model approach enabled us to conduct a thorough comparative analysis between the original English MedNLI dataset and our Turkish translations. To enhance the robustness of our evaluation, we augmented the training process by

incorporating SNLI and MultiNLI datasets, along with their Turkish counterparts. This comprehensive evaluation framework allowed us to investigate whether cross-domain knowledge from different NLI datasets could enhance the overall performance on medical language inference tasks.

For our second evaluation phase, we implemented a systematic human assessment protocol utilizing medical residents with proficiency in both English and Turkish. We developed a comprehensive evaluation survey where medical professionals rated multiple quality aspects on a 10-point scale. Given resource constraints, we employed the Raosoft sampling methodology (*Rao & Rao, 2009*) to determine an optimal sample size. We used a 95% confidence level and a 2% margin of error; as a result, 2,052 examples were carefully selected. While our dataset's overall quality could be assessed with gold labels from NLI relations, the evaluation of abbreviation handling required specialized medical expertise. Our selection of 2,052 examples was strategically driven by the distribution of medical abbreviations in our adapted dataset, ensuring comprehensive coverage of all abbreviation patterns and their translations. To systematically evaluate these examples, we developed a five dimensional evaluation framework focusing on

- Quality of abbreviations derived from sentences.
- Quality of expanded forms of abbreviations.
- Quality of normalized sentences which are reconstructed with abbreviation information along with metric correction.
- The Turkish translations of the normalized sentences.
- The Turkish translations of the expanded forms of the abbreviations.

The resulting statistical analysis, including means and standard deviations across all evaluation dimensions, provided robust validation of both our abbreviation handling methodology and the overall dataset quality.

## RESULTS

Our analysis integrates quantitative performance metrics from state-of-the-art language models BERT, BioBERT, BERTurk, and BioBERTurk with qualitative insights gathered through expert human evaluation by medical residents. We first examine the model-based performance metrics, followed by detailed human evaluation results, and conclude with an analysis of the dataset's strengths and areas for potential enhancement. This multi-faceted evaluation approach provides a thorough understanding of TurkMedNLI's effectiveness in capturing medical language inference patterns while identifying specific opportunities for future improvements.

### Machine evaluation

To create a meaningful baseline for our Turkish translations, we first fine-tuned the original BERT model on English MedNLI sentences. A key innovation in our approach was the integration of extracted abbreviation information to restructure sentences, incorporating both expanded abbreviations and standardized measurements. This

restructuring process effectively served as a normalization step, creating two distinct versions of the NLI pairs for evaluation: the original and the normalized form. To ensure thorough validation, we evaluated BERT's performance on both versions. We further enriched our evaluation by incorporating SNLI and MultiNLI datasets alongside MedNLI to see how the diversity could affect the overall solutions. This comprehensive approach allowed us to assess all aspects of the NLI pipeline while leveraging the benefits of additional high-quality training data.

The results from Table 8 demonstrate the effectiveness of various BERT model configurations across different training combinations and testing scenarios. When trained exclusively on MedNLI, BERT achieves a solid baseline performance with 79.21% training accuracy and 78.19% test accuracy. However, the model's performance on the normalized MedNLI dataset (containing expanded abbreviations and standardized measurements) shows a notable decline to 76.27% training and 72.64% test accuracy, suggesting initial challenges in processing the enhanced contextual information. A significant performance improvement emerges when incorporating SNLI data alongside MedNLI. The BERT + MedNLI + SNLI configuration achieves an impressive 89.03% training accuracy while maintaining a robust 78.76% test accuracy on the original MedNLI dataset. More importantly, this combination shows enhanced performance on the normalized dataset, reaching 80.23% test accuracy. This improvement indicates that the additional linguistic patterns learned from SNLI help the model better understand the expanded contexts.

Further augmentation with MNLI data (BERT + MedNLI + SNLI + MNLI) demonstrates consistent performance, achieving 86.25% training accuracy and maintaining 78.19% test accuracy on the original MedNLI dataset. This configuration shows particular strength on the normalized dataset, reaching 79.25% test accuracy. Notably, models trained exclusively on general-domain datasets (BERT + SNLI, BERT + MNLI, or their combination) show significantly reduced performance when tested on MedNLI, with test accuracies ranging from 60.47% to 62.79%. This performance gap clearly demonstrates that general language understanding, while valuable, is insufficient for medical domain inference tasks. The results emphasize the crucial importance of domain-specific training data for achieving accurate medical language understanding. These findings reveal that while normalization initially poses challenges for the model, the integration of diverse NLI datasets helps overcome these difficulties, ultimately leading to improved performance on both original and normalized medical texts. The results also underscore the value of combining domain-specific and general-domain training data for optimal performance in specialized NLI tasks.

In Table 9, BioBERT demonstrates exceptional performance in medical domain tasks, achieving impressive baseline metrics with 83.51% training accuracy and test accuracies of 82.63% and 83.26%. This performance substantially exceeds standard BERT results, highlighting BioBERT's enhanced capabilities in processing medical text. When examining the model's performance on normalized MedNLI data independently, it maintains strong results with 83.72% training accuracy and 80.30% test accuracy, and 83.40% accuracy on normalized test results, suggesting that BioBERT's biomedical pre-training enables effective processing of detailed medical terminology and expanded abbreviations. The

**Table 8  MedNLI accuracy results with the BERT model**

| Model | Train accuracy | MedNLI test | MedNLI normalized test | SNLI test | MNLI matched | MNLI mismatched |
|---|---|---|---|---|---|---|
| Bert + MedNLI | 79.21 | 78.19 | 78.34 | 38.91 | 48.01 | 49.45 |
| Bert + MedNLI + SNLI | 89.03 | 78.76 | 80.23 | 90.03 | 72.06 | 72.00 |
| Bert + MedNLI + SNLI + MNLI | 86.25 | 78.19 | 79.25 | 90.36 | 82.78 | 82.96 |
| Bert + MedNLI Normalized | 76.27 | 70.25 | 72.64 | 38.43 | 47.09 | 46.61 |
| Bert + MedNLI Normalized + SNLI | 89.57 | 76.93 | 80.09 | 90.57 | 72.62 | 72.96 |
| Bert + MedNLI Normalized + SNLI + MNLI | 86.94 | 78.27 | 80.37 | 90.16 | 83.51 | 83.77 |
| Bert + SNLI | 90.14 | 60.47 | 61.67 | 89.80 | 71.93 | 72.28 |
| Bert + MNLI | 82.61 | 62.79 | 63.50 | 76.12 | 82.35 | 82.97 |
| Bert + SNLI + MNLI | 87.66 | 61.25 | 63.64 | 90.65 | 83.67 | 84.10 |

**Table 9  MedNLI accuracy results with the BioBERT model.**

| Model | Train accuracy | MedNLI test | MedNLI normalized test | SNLI test | MNLI matched | MNLI mismatched |
|---|---|---|---|---|---|---|
| BioBert + MedNLI | 83.51 | 82.63 | 83.26 | 43.17 | 53.79 | 55.03 |
| BioBert + MedNLI + SNLI | 88.11 | 82.34 | 82.77 | 88.75 | 72.50 | 73.18 |
| BioBert + MedNLI + SNLI + MNLI | 85.41 | 82.20 | 83.12 | 88.68 | 82.07 | 83.45 |
| BioBert + MedNLI Normalized | 83.72 | 80.30 | 83.40 | 42.13 | 50.66 | 51.68 |
| BioBert + MedNLI Normalized + SNLI | 88.91 | 81.50 | 84.88 | 89.03 | 72.90 | 73.44 |
| BioBert + MedNLI Normalized + SNLI + MNLI | 86.18 | 81.22 | 84.31 | 89.33 | 82.42 | 83.78 |
| BioBert + SNLI | 89.62 | 68.42 | 71.87 | 88.94 | 72.49 | 73.59 |
| BioBert + MNLI | 82.46 | 68.77 | 71.80 | 74.37 | 82.05 | 83.27 |
| BioBert + SNLI + MNLI | 85.53 | 69.83 | 72.85 | 88.81 | 81.97 | 82.90 |

integration of SNLI data with MedNLI enhances BioBERT's capabilities, with the BioBERT + MedNLI + SNLI configuration achieving 88.11% training accuracy and 82.34% test accuracy on the original dataset. When this same configuration is applied to normalized data (BioBERT + MedNLI Normalized + SNLI), it achieves 81.50% test accuracy and 84.88% normalized test accuracy. These results indicate that combining medical domain knowledge with general language patterns helps maintain strong performance across both original and normalized medical texts. The addition of MNLI data to create the complete configuration (BioBERT + MedNLI + SNLI + MNLI) demonstrates consistent performance, achieving 82.20% test accuracy on the original dataset and improving to 83.12% on the normalized version. Models trained exclusively on general-domain datasets (BioBERT + SNLI, BioBERT + MNLI, or their combination) show notably lower performance on MedNLI, with test accuracies ranging from 68.42% to 71.87%. However, these results still surpass standard BERT's performance on similar configurations, indicating BioBERT's inherent advantage in medical domain tasks.

**Table 10 TurkMedNLI accuracy results with the BERTurk model.**

| Model | Train accuracy | MedNLI test | MedNLI normalized test | SNLI-Tr test | MNLI-Tr matched | MNLI-Tr mismatched |
|---|---|---|---|---|---|---|
| BertTurk + TurkMedNLI | 76.20 | 75.17 | 76.44 | 36.95 | 50.47 | 51.18 |
| BertTurk + TurkMedNLI + SNLI-Tr | 86.53 | 75.94 | 76.86 | 87.22 | 69.06 | 69.56 |
| BertTurk + TurkMedNLI + SNLI-Tr + MNLI-Tr | 83.25 | 77.98 | 79.32 | 87.57 | 79.38 | 79.68 |
| BertTurk + TurkMedNLI Normalized | 78.99 | 74.05 | 76.30 | 38.17 | 50.47 | 51.65 |
| BertTurk + TurkMedNLI Normalized + SNLI-Tr | 86.81 | 75.38 | 78.34 | 87.66 | 69.49 | 70.28 |
| BertTurk + TurkMedNLI Normalized + SNLI-Tr + MNLI-Tr | 83.12 | 77.63 | 78.55 | 87.10 | 79.01 | 79.82 |
| BertTurk + SNLI-Tr | 87.29 | 60.47 | 60.82 | 87.48 | 69.83 | 70.71 |
| BertTurk + MNLI-Tr | 79.22 | 60.33 | 63.22 | 72.88 | 78.98 | 80.56 |
| BertTurk + SNLI-Tr + MNLI-Tr | 83.22 | 62.09 | 64.20 | 87.36 | 79.24 | 80.63 |

The Table 10 shows BERTurk's performance on Turkish language medical inference task and demonstrates both the challenges and opportunities in cross-lingual medical NLP. When trained exclusively on TurkMedNLI, the model achieves a baseline performance of 76.20% training accuracy and 75.17% test accuracy on the original dataset. The model's interaction with normalized data shows an interesting pattern, achieving slightly higher training accuracy of 78.99% and test accuracy of 76.30%. This initial performance suggests that BERTurk not only effectively processes Turkish medical text, but actually shows improved performance when handling normalized medical terminology, indicating that the standardization of medical terms may facilitate better understanding in the Turkish language context. The integration of SNLI-Tr data with TurkMedNLI yields notable improvements in the model's capabilities. The BERTurk + TurkMedNLI + SNLI-Tr configuration achieves 86.53% training accuracy and 75.94% test accuracy on the original dataset. When this same configuration is tested on the normalized dataset (BERTurk + TurkMedNLI Normalized + SNLI-Tr), it shows improved performance with 78.34% test accuracy. This enhancement in performance suggests that exposure to additional Turkish language patterns from SNLI-Tr helps the model better process both standard and normalized medical text.

Further augmentation with MNLI-Tr data demonstrates the value of comprehensive training data. The complete configuration (BERTurk + TurkMedNLI + SNLI-Tr + MNLI-Tr) achieves 77.98% test accuracy on the original dataset and maintains strong performance. When the same process is repeated with the normalized dataset, we achieve a slightly higher 78.55% test accuracy on the normalized version. This consistent performance across both original and normalized datasets indicates that the combination of medical and general language understanding in Turkish creates a more robust model for medical inference tasks. The performance pattern of models trained exclusively on general domain datasets provides important insights: BERTurk + SNLI-Tr, BERTurk + MNLI-Tr, and their combinations show significantly reduced performance on TurkMedNLI, with test accuracies ranging from 60.33% to 62.09%. These results align with patterns observed

**Table 11  TurkMedNLI accuracy results with BioBERTurk model.**

| Model | Train accuracy | MedNLI test | MedNLI normalized test | SNLI-Tr test | MNLI-Tr matched | MNLI-Tr mismatched |
|---|---|---|---|---|---|---|
| BioBertTurk + TurkMedNLI | 77.63 | 75.59 | 76.23 | 37.92 | 46.09 | 47.24 |
| BioBertTurk + TurkMedNLI + SNLI-Tr | 83.39 | 75.24 | 75.31 | 83.92 | 62.47 | 63.35 |
| BioBertTurk + TurkMedNLI + SNLI-Tr + MNLI-Tr | 79.86 | 75.24 | 76.44 | 84.34 | 75.66 | 76.34 |
| BioBertTurk + TurkMedNLI Normalized | 74.12 | 71.72 | 72.29 | 37.98 | 41.10 | 41.74 |
| BioBertTurk + TurkMedNLI Normalized + SNLI-Tr | 81.84 | 71.94 | 73.41 | 82.45 | 62.09 | 63.66 |
| BioBertTurk + TurkMedNLI Normalized + SNLI-Tr + MNLI-Tr | 80.01 | 76.37 | 72.92 | 84.46 | 75.62 | 76.25 |
| BioBertTurk + SNLI-Tr | 83.04 | 57.24 | 56.82 | 82.85 | 61.44 | 62.38 |
| BioBertTurk + MNLI-Tr | 74.77 | 60.12 | 62.37 | 63.20 | 74.08 | 76.28 |
| BioBertTurk + SNLI-Tr + MNLI-Tr | 79.18 | 60.97 | 63.99 | 83.50 | 74.84 | 76.21 |

in English BERT models, suggesting that the challenges of medical domain adaptation are consistent across languages.

The Table 11 presents BioBERTurk's performance on the Turkish medical inference task, revealing interesting patterns in the intersection of domain-specific knowledge and language-specific challenges. When trained exclusively on TurkMedNLI, the model achieves a baseline performance of 77.63% training accuracy and 75.59% test accuracy. However, when processing normalized data, the performance shows a notable decline with 74.12% training accuracy and 72.29% test accuracy. This initial performance pattern suggests that while BioBERTurk effectively handles basic Turkish medical text, the additional complexity introduced by normalized medical terminology presents specific challenges that warrant further investigation. The integration of SNLI-Tr data with TurkMedNLI demonstrates the model's potential for enhancement through diverse training data. The BioBERTurk + TurkMedNLI + SNLI-Tr configuration achieves 83.39% training accuracy and 75.24% test accuracy on the original dataset. When evaluated on the normalized dataset, this configuration maintains relatively stable performance with 73.41% test accuracy. This pattern indicates that while additional Turkish language patterns from SNLI-Tr contribute to the model's overall capabilities, the challenges of processing normalized medical text persist.

The addition of MNLI-Tr data to create the complete configuration (BioBERTurk + TurkMedNLI + SNLI-Tr + MNLI-Tr) shows mixed results in the model's performance. While this configuration achieves 75.24% test accuracy on the original dataset, the performance decreases to 72.92% when using the normalized dataset configuration. These results indicate that while the combination of multiple datasets can enhance the model's general capabilities, the handling of normalized medical terminology presents particular challenges that require further optimization. The performance trends across different configurations highlight both the strengths and areas for improvement in BioBERTurk's medical language understanding capabilities. Models trained exclusively on general

domain datasets show significantly reduced performance, with test accuracies ranging from 57.24% to 63.99%. These findings indicate that while the current implementation successfully adapts biomedical knowledge to Turkish language structures, there remains an opportunity for enhancement in processing normalized medical content through refined training approaches and architectural adjustments.

## Medical resident evaluation

We have evaluated our translation accuracy and abbreviation extraction quality with medical residents. Using Raosoft, we extracted 2,052 rows from TurkMedNLI and divided this into three groups. Each group evaluated five different aspects, ranging from abbreviation extraction to normalized translation quality. First, we took the mean and standard deviation of each evaluator's results as presented in Table 12.

**Evaluation of abbreviation quality:** The quality of abbreviations indicates that each demonstrates high satisfaction, but with varying levels of consistency. Resident-1 reported a high mean satisfaction score of 9.821 with a low standard deviation of 0.820, indicating consistent positive evaluations. Resident-2's mean score was slightly lower at 9.75, with a standard deviation of 1.003, suggesting a broader range of scores. Resident-3's evaluations showed a mean score of 9.436 with a higher standard deviation of 1.437, reflecting the most significant variability and the lowest average satisfaction among the three residents.

**Evaluation of abbreviation expansion quality:** The quality of extended abbreviations indicates overall satisfaction, but with varying levels of consistency. Resident-1 reported a mean score of 8.805 with a standard deviation of 2.191, indicating moderate variability in evaluations. Resident-2's mean score was slightly higher at 8.910 with a standard deviation of 2.292, suggesting more consistent evaluations than Resident-1. Resident-3's evaluations showed a mean score of 8.923 with a standard deviation of 2.330, reflecting the most consistent evaluations among the three residents, though slightly lower in satisfaction compared to the others.

**Evaluation of expanded abbreviation translation:** The quality of expanded abbreviation translations demonstrates varying levels of satisfaction and consistency. Resident-1 reported a mean score of 8.251 with a standard deviation of 2.401, indicating relatively excellent quality with moderate variability. Resident-2's mean score was lower at 7.930 with a higher standard deviation of 2.629, suggesting moderate satisfaction and higher variability. Resident-3's evaluations showed a mean score of 9.201 with a lower standard deviation of 1.827, reflecting high satisfaction and more consistent evaluations compared to the other residents.

**Evaluation of normalized sentence quality:** The quality of normalized sentences indicates high satisfaction, with varying levels of consistency. Resident-1 reported a mean score of 8.976 with a standard deviation of 2.089, indicating high satisfaction with low variability. Resident-2's mean score was slightly higher at 9.074 with a standard deviation of 1.725, suggesting high satisfaction with slightly lower variability. Resident-3's evaluations showed a mean score of 9.018 with a standard deviation of 2.123, reflecting consistent high satisfaction with moderate variability.

**Table 12  Evaluation metrics for TurkMedNLI per evaluator.**

| Metric | Evaluator-1 | Evaluator-2 | Evaluator-3 |
|---|---|---|---|
| Abbreviation quality mean | 9.821256 | 9.750000 | 9.435691 |
| Abbreviation quality std | 0.820267 | 1.003064 | 1.437326 |
| Abbreviation expanded quality mean | 8.805153 | 8.910131 | 8.922830 |
| Abbreviation expanded quality std | 2.190678 | 2.292112 | 2.330328 |
| Abbreviation expanded translation quality mean | 8.251208 | 7.929739 | 9.200965 |
| Abbreviation expanded translation quality std | 2.401213 | 2.628678 | 1.827452 |
| Normalized sentence quality mean | 8.975845 | 9.073529 | 9.017685 |
| Normalized sentence quality std | 2.088616 | 1.724803 | 2.122954 |
| Normalized sentence translation quality mean | 7.067633 | 7.488562 | 7.770096 |
| Normalized sentence translation quality std | 2.039592 | 2.124951 | 2.404412 |

**Evaluation of normalized sentence translation quality:** Normalized sentence translation quality indicates moderate satisfaction with varying consistency. Resident-1 reported a mean score of 7.068 with a standard deviation of 2.040, indicating moderate satisfaction with moderate variability. Resident-2's mean score was slightly higher at 7.489 with a standard deviation of 2.125, suggesting slightly higher satisfaction with similar variability. Resident-3's evaluations showed a mean score of 7.770 with a standard deviation of 2.404, reflecting the highest satisfaction among the three residents with moderate variability. A critical consideration in evaluating our Turkish medical translation quality is the validation of expert assessments across medical residents. Given our dataset's extensive size and the time constraints of medical experts, we implemented a strategic sampling approach where each resident evaluated a distinct subset of translations. This methodological choice, while allowing broader coverage of translations, presents a fundamental limitation in assessing inter-rater reliability. Traditional reliability measures such as Cohen's Kappa or Intraclass Correlation Coefficient require multiple evaluators to assess identical items to measure direct agreement. In our study design, each medical resident evaluated entirely different sets of translations, with no overlap between the assessed items, making it mathematically challenging to calculate rating agreement, as these metrics require paired comparisons of how different evaluators scored the exact same translations.

While statistical approaches exist for comparing rating distributions across evaluators, such as Kruskal-Wallis tests, applying these methods to our non-overlapping evaluation sets could lead to misleading conclusions, as any observed differences might be attributed to variations in the inherent difficulty of the evaluated translations rather than genuine disagreement between residents. Nevertheless, our descriptive statistics reveal consistently high mean ratings across evaluators for all evaluation criteria. However, we acknowledge that this observed consistency cannot be statistically validated due to our study design's constraints. This limitation originated from prioritizing comprehensive coverage over the ability to measure direct rating agreement, a trade-off necessitated by the practical constraints of medical expert availability and dataset size. Future research would benefit

from incorporating a subset of overlapping evaluations while maintaining the coverage achieved through our current approach.

The residents' evaluation of the dataset reveals high satisfaction levels in a variety of areas, including high scores for abbreviation quality and sentence normalization. However, there is variation in the extended abbreviation and translation quality, indicating areas for improvement. The high ratings for normalized sentence quality suggest that the normalization process enhanced the clarity and comprehensibility of the sentences. However, translations of normalized sentences received moderate scores, implying that more fine-tuning may be required for more consistent outcomes. The evaluations highlight the dataset's strengths in abbreviation handling and sentence normalization while also identifying areas for improvement, particularly in translation. These findings inform future improvements to the dataset's quality and reliability for medical language inference tasks. Residents' feedback validates the current approach's effectiveness and provides a clear path for targeted improvement.

Figure 2 presents a comprehensive visualization of the distribution of evaluation rankings for various quality metrics assessed by three medical residents. The metrics evaluated abbreviation quality, extended abbreviation quality, extended abbreviation translation quality, normalized sentence quality, and normalized sentence translation quality. The plots represent the frequency of rankings from 1 to 10, reflecting the level of satisfaction and consistency in the evaluations.

Abbreviation quality shows an overwhelmingly high level of quality among the residents, with the vast majority of rankings at 10. The distribution for extended abbreviation quality shows a slightly broader range of rankings, though still predominantly high. The plot for extended abbreviation translation quality exhibits a wider distribution of rankings compared to the previous metrics, suggesting moderate satisfaction with translation quality and the need for refinement to enhance consistency and overall satisfaction.

The plot for normalized sentence quality shows a strong concentration of high rankings, indicating high satisfaction with the quality of normalized sentences. The distribution for normalized sentence translation quality is more varied, with rankings spread across the scale, suggesting moderate satisfaction with the translation process. The consistent high rankings for abbreviation and normalized sentence quality validate the effectiveness of these aspects, while the distribution of lower rankings in translation-related metrics highlights opportunities for improvement. These insights are crucial for guiding future enhancements to the dataset, ensuring higher consistency and overall satisfaction in medical language inference tasks.

## DISCUSSIONS

The performance decrease between English and Turkish translations in our results originates from the limitations of our translation model. Facebook's NLLB model is great at processing multiple languages simultaneously. However, its training primarily relies on general-purpose parallel corpora. To address these limitations, future research could focus on developing a domain-specific English-Turkish parallel medical *corpus*. Although this

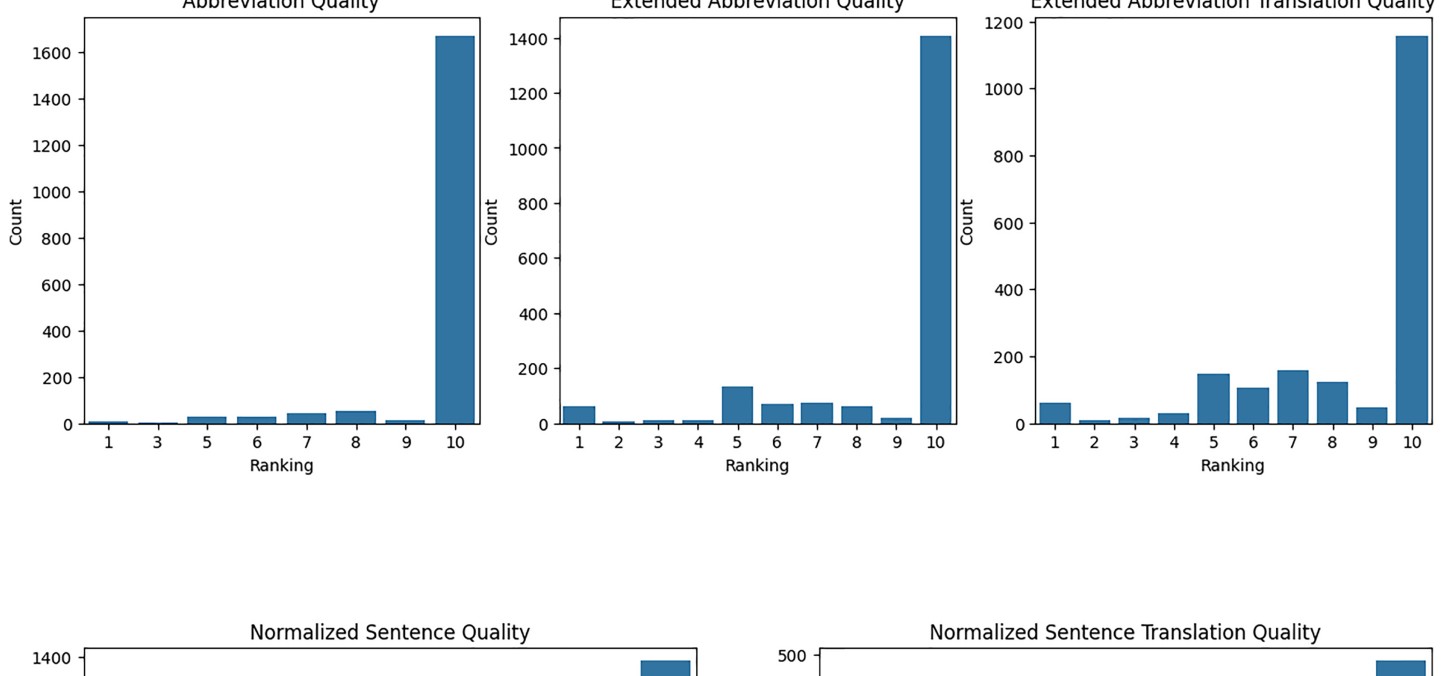

**Figure 2** TurkMedNLI evaluation distribution per evaluation criteria.

solution demands substantial data curation resources, it presents an effective pathway for improving translation quality and reducing performance disparities between languages.

While TurkMedNLI demonstrates the effectiveness of using large language models for medical dataset translation and refinement, there are several important limitations. The computational costs associated with LLM fine-tuning present a significant constraint, which requires carefully selecting a subset of training data to make the abbreviation extraction model training feasible. The human evaluation component of our pipeline presents additional scalability challenges. As dataset size increases or medical domain coverage expands, the need for domain experts with diverse specializations grows proportionally, increasing both validation costs and potential bottlenecks in the dataset creation process.

TurkMedNLI builds upon the foundational work established by RuMedNLI and ViMedNLI while introducing a novel approach to domain-specific translation challenges. RuMedNLI demonstrated the effectiveness of combining machine translation with expert

human validation, achieving 77.64% accuracy with their RuBERT model on medical inference tasks. ViMedNLI advanced this methodology through state-of-the-art English-Vietnamese translation systems, with their ViPubmedT5 model reaching 81.65% accuracy and ViHealthBERT achieving 79.04% accuracy. In comparison, our BioBERTurk model achieves 75.59% accuracy on direct translation and 72.99% on the normalized version, which, while slightly lower than its predecessors, represents comparable performance considering Turkish's complex morphological structure and the challenges of medical domain adaptation.

## CONCLUSION

In this study, we introduced TurkMedNLI, the first Turkish Medical Natural Language Inference dataset, developed using a large language model-based pipeline. Our approach successfully addresses critical challenges in domain-specific translation, particularly the disambiguation of medical abbreviations, measurement and metric conversion, and preservation of clinical context. The pipeline's effectiveness, demonstrated through both machine learning metrics and expert validation, establishes a framework for creating high-quality medical datasets in low-resource languages.

In addition to its contribution to NLI tasks, the dataset offers valuable resources for broader applications in biomedical NLI. The parallel *corpus* of medical abbreviations and their expanded forms, along with contextual sentences, can significantly enhance NER models' ability to identify and process domain-specific entities. These improvements have broader implications for clinical text mining, automated medical information extraction, and cross-lingual medical data applications.

Despite challenges, our study demonstrates the broad potential of large language models in creating high-quality domain-specific datasets. Our methodology for handling specialized terminology and context-aware translation extends beyond Turkish medical NLP, offering a framework adaptable to other specialized domains and low-resource languages.

Looking ahead, we have identified several key opportunities for advancing this research. Our first priority is to enhance the preprocessing methodology to handle a wider range of medical examples. We also aim to develop specialized English-Turkish parallel corpora for medical texts, which will enable more precise domain-specific translation model fine-tuning. In parallel, we plan to explore smaller, more efficient language models that can be comprehensively fine-tuned with larger datasets. A critical focus will be implementing a more rigorous medical expert evaluation framework that includes overlapping assessment sets for proper statistical validation of inter-rater reliability and standardized evaluation metrics across different medical specialties.

## ACKNOWLEDGEMENTS

We are deeply grateful to the medical professionals who generously contributed their expertise to verify the translations for this study. Special thanks go to Dr. Gonca Bakir, an ophthalmology resident; Dr. Tarık İnci, an internal medicine resident; and Dr. İrem Şevik, a public health resident. Their invaluable insights, meticulous reviews, and unwavering

commitment have greatly improved the quality and accuracy of our work. We are grateful for their support and dedication to this project, which has contributed significantly to its success.

### Funding
The authors received no funding for this work.

### Competing Interests
The authors declare that they have no competing interests.

### Author Contributions
- İskender Ülgen Oğul conceived and designed the experiments, performed the experiments, analyzed the data, performed the computation work, prepared figures and/or tables, authored or reviewed drafts of the article, and approved the final draft.
- Fatih Soygazi conceived and designed the experiments, performed the experiments, analyzed the data, performed the computation work, prepared figures and/or tables, authored or reviewed drafts of the article, and approved the final draft.
- Belgin Ergenç Bostanoğlu conceived and designed the experiments, performed the experiments, analyzed the data, performed the computation work, prepared figures and/or tables, authored or reviewed drafts of the article, and approved the final draft.

### Data Availability
The code is available in the Supplemental File.

The third party data is available at https://physionet.org/content/mednli/1.0.0. Access is limited to credentialed users who sign the DUA.

### Supplemental Information
Supplemental information for this article can be found online at http://dx.doi.org/10.7717/peerj-cs.2662#supplemental-information.

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
