# Peer review of "TurkMedNLI: a Turkish medical natural language inference dataset through large language model based translation"

_PeerJ Computer Science, doi:10.7717/peerj-cs.2662_

## Round 0.1 · original submission · Major Revisions

Dear Authors,

Thank you for submitting your article. Based on reviews' comments, your article has not yet been recommended for publication in its current form. However, we encourage you to address the concerns and criticisms of the reviewer and to resubmit your article once you have updated it accordingly. Furthermore, equations should be used with correct equation number. Please do not use “as follows”, “given as”, etc. Explanation of the equations should also be checked. All variables should be written in italic as in the equations. Their definitions and boundaries should be defined. Necessary references should be provided. Many of the equations are part of the related sentences. Attention is needed for correct sentence formation.

Best wishes,

·

Basic reporting

The authors present TurkMedNLI, a Turkish medical natural language inference (NLI) dataset developed through a novel pipeline combining large language models (LLMs) for translation and sentence reconstruction. The dataset addresses the gap in resources for Turkish medical NLP, a low-resource language, and offers solutions to translation challenges, such as abbreviations and metric conversions. While the study introduces valuable contributions to Turkish NLP research, several areas require major revisions to enhance clarity, methodology, and overall impact.

1. The manuscript is generally well-written, but there are sections where clarity could be improved. For instance, certain complex sentences could benefit from simplification. The introduction is informative but lacks a smooth transition between the general concept of NLI and the specific challenges of medical NLI. This disjointedness may confuse readers who are less familiar with the topic. Consider revising for flow and readability.

2. The authors adequately review related work, especially in the area of low-resource NLI datasets and medical NLP. However, more recent studies on multilingual medical NLP datasets could strengthen the argument. Additionally, some comparisons to RuMedNLI and ViMedNLI should be more explicit regarding performance differences. Expand the literature review to include more recent research and clarify comparisons between your method and prior work.

3. The figures and tables provide helpful insights, especially regarding the performance metrics. However, some figures (e.g., Figure 2) are cluttered and could be more visually appealing. Additionally, several tables lack a detailed explanation of their content. Improve the clarity of figures and provide more detailed captions for the tables to ensure that they are self-explanatory.

Experimental design

1. The design of the translation pipeline using Llama-3.1 and NLLB models is innovative. However, the rationale behind certain decisions (e.g., choice of abbreviation dataset or preprocessing steps) is not fully explained. For example, how were the abbreviation density thresholds determined, and why were specific LLMs chosen over others? Furthermore, the decision to use pre-trained models like BERT and BioBERT instead of more advanced models could be reconsidered given recent advancements in transformer models. Provide more detailed justifications for the selection of models, thresholds, and other methodological choices. Consider evaluating more advanced models beyond BERT/BioBERT for comparison.

2. The human evaluation conducted by medical experts is valuable but somewhat limited in scope. The manuscript lacks details on the specific criteria used by experts and whether inter-rater reliability (e.g., Cohen’s Kappa) was assessed to ensure consistency across evaluators. Expand on the evaluation methodology, including details on inter-rater reliability and the selection of examples for evaluation.

Validity of the findings

1. The results, particularly with BERT and BioBERT models, are encouraging. However, the performance drop in Turkish translations compared to English models raises concerns. The authors attribute this to translation complexity, but this explanation could be more rigorously examined. Explore in more detail why the performance drops for Turkish translations and whether any adjustments to the pipeline could mitigate this. For instance, would further fine-tuning of models specifically for Turkish medical language improve results?

2. The conclusion highlights the contributions of the TurkMedNLI dataset but underplays its limitations, especially regarding the scalability of the proposed approach for future datasets. Additionally, the novelty claim regarding handling metric conversions and abbreviations could be better substantiated by comparisons with previous studies. Revisit the conclusion to more explicitly address limitations and areas for future improvement. Provide a more thorough comparison with related work to justify novelty claims.

Additional comments

While the manuscript offers valuable contributions to Turkish medical NLP, several revisions are necessary to improve its clarity and rigor. Enhancing the methodological justification, expanding human evaluation, and addressing performance discrepancies will significantly strengthen the study.

Reviewer 2 ·

Basic reporting

Please check the comments in additional comments.

Experimental design

Please check the comments in additional comments.

Validity of the findings

Please check the comments in additional comments.

Additional comments

1. Some minor improvements are required in the abstract. It is lengthy, so I suggest decreasing the background information in the abstract. In lines 34 and 35, the authors have written “76,23% and 76,44%” which should be changed to 76.23% and 76.44%.
2. There are inconsistencies in the way the abbreviations are written. In line 64, the author does not capitalize the full form of the abbreviation.
3. Line 67, “Facebook’s No Language Left Behind,” is ambiguous. I suggest you write it in italics so it is easily readable.
4. In line 100 “Inference in Turkish (Budur et al., 2020) study” should be re-written.
5. In line 125, the author explains that the dataset is available. I suggest the author add the link to the dataset in the footnote.
6. In Line 149, two references should be written together “(Romanov and Shivade, 2018, Goldberger et al., 2000)” and instead of giving a reference at the end of the paragraph I suggest you give it in the first line where the two studies are discussed.
7. Tables and Figures should be written as “Table 1” not “Table-1”.
8. In the technical background, the authors should use past tense instead of present tense.
9. On line 269 “as noted by (Budach et al., 2022)” the reference should not be in brackets as it is being read in the sentence.
10. Mostly paragraphs end with references, I suggest you add references at the start of the study being discussed – at the end of the sentence where the discussion about the paper appears.
11. Figures 1 and 4 are not clear. In Figure 4, the authors have not explained the terms and also what does the plus sign indicates – its not clear.

---

## Round 0.2 · accepted · Accept

Dear Authors,

Thank you for clearly addresing the reviewers' comments. Your manuscript now seems sufficiently improved and ready for publication.

Best wishes,

·

Basic reporting

The authors addressed all my concerns. The paper can now be accepted for publication.

Experimental design

The authors addressed all my concerns. The paper can now be accepted for publication.

Validity of the findings

The authors addressed all my concerns. The paper can now be accepted for publication.

Additional comments

The authors addressed all my concerns. The paper can now be accepted for publication.

Reviewer 2 ·

Basic reporting

Check additional comments.

Experimental design

Check additional comments.

Validity of the findings

Check additional comments.

Additional comments

The authors have incorporated all the changes as previously suggested. I have no further suggestions about the paper.